# Is social connectedness a risk factor for the spreading of COVID-19 among older adults? The Italian paradox

**Giuseppe Liotta** [1]☯*, **Maria Cristina Marazzi**[2]☯, **Stefano Orlando**[1]☯, **Leonardo Palombi**[1]☯

**1** Biomedicine and Prevention Dept., University of Rome "Tor Vergata", Rome, Italy, **2** LUMSA University, Rome, Italy

☯ These authors contributed equally to this work.
* Giuseppe.liotta@uniroma2.it

**Data Availability Statement:** All relevant data are within the manuscript.

**Funding:** the authors received no specific funding for this work.

## Abstract

Italy was one of the first European countries affected by the new coronavirus (COVID-19) pandemic, with over 105,000 infected people and close to 13,000 deaths, until March 31st. The pandemic has hit especially hard because of the country's demographic structure, with a high percentage of older adults. The authors explore the possibility, recently aired in some studies, of extensive intergenerational contact as a possible determinant of the severity of the pandemic among the older Italian adults. We analyzed several variables to test this hypothesis, such as the percentage of infected patients aged >80 years, available nursing home beds, COVID-19 incidence rate, and the number of days from when the number of positive tests exceeded 50 (epidemic maturity). We also included in the analysis mean household size and percentage of households comprising one person, in the region. Paradoxically, the results are opposite of what was previously reported. The pandemic was more severe in regions with higher family fragmentation and increased availability of residential health facilities.

## Introduction

Between December 31, 2019 and March 31, 2020, 750,890 cases and 36,405 deaths due to new coronavirus (COVID-19) have been reported [1]. As is well known, Italy is one of the countries most affected by the pandemic, with 12,428 deaths and 105,792 cases recorded in the months of February and March 2020 [2]. Many have wondered why COVID-19 has hit so hard in Italy, but unfortunately, over the weeks it has become increasingly clear that other EU countries as well as the US are developing similar growth trends. Recently some studies [3] stated that age, in both local and national context, as well as the social connectedness of older and younger generations act as powerful determinants in the spread of pandemics. While there is a very clear association between the case fatality rate and age demographics (Italy has the second oldest population worldwide and has the highest ageing index in Europe [4] with a value of 168.9), we wanted to test the hypothesis that the supposed closeness between younger and older generations in Italian families may have played a major role in the pandemic spread. In

**Competing interests:** the authors have declared taht no competing interest exists.

fact, Dowd, et al. stated that "Italy is also a country characterized by extensive intergenerational contacts which are supported by a high degree of residential proximity between adult children and their parents." [3]. According to this observation, we expected to find a strong direct association between family size and the spread of COVID-19. Conversely, social distance should be favored in cities and regions with a high number of residential structures for long term care, usually devoted to older adults.

## Methods

The study is based on the available population data from each Italian administrative region. The sources of data used in this paper are the daily situation reports on COVID-19 published by the Italian Ministry of Health (28 February– 31 march), the national Institute of STATistics (ISTAT) 2019 data set on households and population, and the 6^th Report generated by the Non-self sufficiency network (2017–2018). In Italy, people aged >80 are 4.33 millions, of which 37.1% are men (mean age 84.7) and 62.9% are women (mean age 85.7). Several variables with plausible association with the spread of COVID-19 among citizens aged >80 years have been taken into consideration. These include the percentage the Italian population aged >80 years [4], COVID-19 incidence rate, and number of days from when the number of confirmed cases exceeded 50 (epidemic maturity)[1]. Additionally, we explored the relationship between the proportion of infected patients aged >80 years and social connectedness indicators, such as the percentage of family comprising one members and household size [5]. Finally, we considered the availability of beds in nursing homes [6] since the pandemic severely impacted nursing home residents throughout the country caused by difficulties in social distancing in these settings as well as the well documented lack of Personal Protective Equipment (PPE) during the first phase of the pandemic in Italy. The population aged >80 years was chosen because they account for about 50% of the deaths due to COVID-19 [7], while representing 7.2%[4]) of the total population and about 75% [8] of the nursing home residents. Univariate (Pearson correlation) and multivariate (linear regression) analyses were conducted to test the hypotheses.

## Results

Table 1 reports the proportion of residents aged >80 years among the total number of COVID-19 cases, according to the Italian Administrative Regions, that ranges from 4.3% for Basilicata to 23.6% for Marche region [9].

The deviation from the Italian average (18.8%) is +25% for the highest value and -75.5% for the lowest. This cannot be explained by the different percentages of residents aged >80 years among the population that ranges from 7.8 to 11.5%. The maturity of the epidemic (the number of days since the infection exceeded 50 confirmed cases) ranges from 10 to 32 days according to region. This could partially explain the differences, even if other factors seem to be involved. The incidence rate among the general population ranged from 4.09/1,000 residents (Valle d'Aosta) to 0.27/1,000 residents (Sicily and Campania) and could explain the spread of the infection among the residents aged >80 years. The mean household size showed a negative correlation with the proportion of infected residents aged >80 years, whereas the percentage of households with one member and availability of nursing homes beds showed a positive correlation with the proportion of infected residents aged >80 years (Table 1).

Multivariate analyses allowed the comparison of different models of the diffusion of the epidemic to the target population (Table 2).

The first model included the percentage of the general population aged >80 years and the measurements of the impact of the pandemic at population level. The incidence rate dropped

**Table 1. Main variables included in the analysis (all data have been extracted between 1st and 7th April 2020).**

| Region | Percentage of aged >80 among COVID-19 cases | COVID-19 Incidence Rate (‰) | Days after achieving 50 cases | Mean Number of households members | Percentage of households with one member | Nursing home beds rate (%) |
|---|---|---|---|---|---|---|
| Piemonte | 19.3 | 1.76 | 27 | 2.18 | 34.4 | 4.1 |
| Valle d'aosta | 18.2 | 4.09 | 15 | 2.08 | 39.6 | 3.7 |
| Lombardia | 19.6 | 3.92 | 32 | 2.26 | 32 | 2.9 |
| Bolzano | 20.5 | 2.09 | 18 | 2.35 | 33.8 | 4.4 |
| Trento | 22.3 | 2.78 | 19 | 2.35 | 34.1 | 4.4 |
| Veneto | 17.3 | 1.62 | 31 | 2.39 | 29.5 | 3.2 |
| Friuli | 19.9 | 1.18 | 21 | 2.17 | 35.6 | 3.2 |
| Liguria | 23.3 | 1.82 | 22 | 2.02 | 40.9 | 2.7 |
| Emilia | 19.9 | 2.78 | 30 | 2.22 | 34.4 | 3 |
| Toscana | 15 | 1.02 | 24 | 2.28 | 32 | 2 |
| Umbria | 8 | 1.1 | 17 | 2.34 | 31.4 | 1.3 |
| Marche | 23.6 | 2.21 | 26 | 2.4 | 29.3 | 2.2 |
| Lazio | 16 | 0.43 | 23 | 2.21 | 34.4 | 1.3 |
| Abruzzo | 12.3 | 0.86 | 17 | 2.38 | 29.6 | 1.3 |
| Molise | 16.5 | 0.4 | 10 | 2.39 | 31.6 | 2 |
| Campania | 8.8 | 0.27 | 23 | 2.72 | 23.6 | 0.7 |
| Puglia | 15 | 0.36 | 20 | 2.58 | 24.8 | 1.2 |
| Basilicata | 4.3 | 0.32 | 10 | 2.48 | 29.6 | 1.5 |
| Calabria | 9 | 0.29 | 16 | 2.49 | 29.7 | 1 |
| Sicilia | 11.1 | 0.27 | 21 | 2.5 | 28.5 | 1.4 |
| Sardegna | 13.3 | 0.38 | 15 | 2.33 | 31.8 | 1.7 |
| Pearson Corr | 0.609 | 0.656 | 0.477 | -0.602 | 0.570 | 0.738 |
| p | 0.003 | 0.001 | 0.029 | 0.004 | 0.007 | <0.001 |

from the stepwise model, showing that the duration of the circulation of the virus among the population, rather than the incidence rate, determined the increased proportion of infections among the population aged >80 years. The second step was to add variables related to social connectedness into the model. At this stage, the percentage of households with only one

**Table 2. Multivariable linear regression (forward stepwise); outcome variable: Percentage of over-80 residents among COVID-19 cases according to Italian administrative regions.** Forward stepwise linear regression.

| | Adjusted $R^2$ (Stat Sign) | Variable | β coeff (Stat. sign) |
|---|---|---|---|
| **Model 1** | 0.559 (0.073) | % of aged>80 on the total population | 0.613 (0.001) |
| | | Days after achieving 50 cases | 0.481 (0.005) |
| | | COVID-19 incidence rate | 0.323 (0.073) |
| **Model 2** | 0.693 (0.039) | % of aged>80 on the total population | 0.480 (0.004) |
| | | Days after achieving 50 cases | 0.418 (0.007) |
| | | % of households comprising one member | 0.116 (0.702) |
| | | Mean number of households members | -0.335 (0.039) |
| **Model 3** | 0.695 (<0.001) | % of aged>80 on the total population | 0.373 (0.021) |
| | | Days after achieving 50 cases | 0.354 (0.015) |
| | | Percentage of Nursing homes beds Combined with the mean number of households members | 0.460 (0.008) |

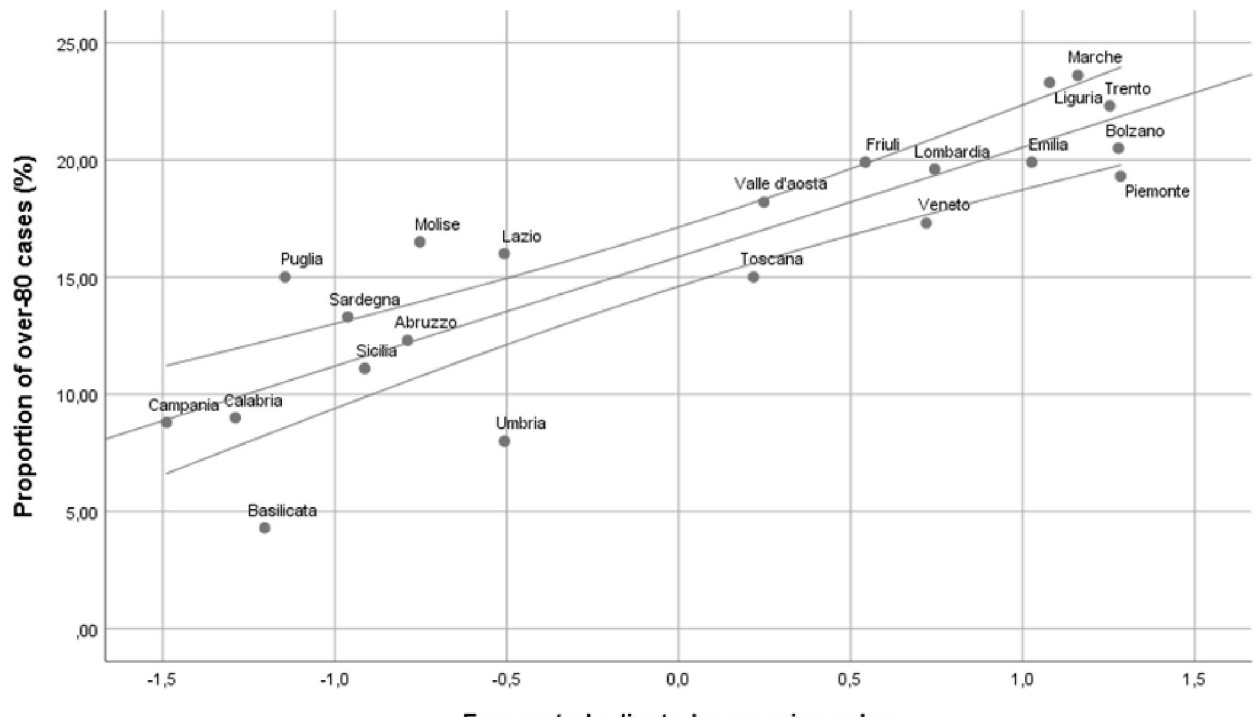

**Fig 1. Proportion of COVID-19 cases generated by residents aged >80 years on total number of cases, according to Italian administrative regions, adjusted for the percentage of residents aged >80 years on the total population, the number of days after achieving 50 cases at regional level and the nursing home beds rate combined with the mean number of households members (adjusted R$^2$: 0.706; F-change: 16.988; Stat Sign <0.001).**

member was excluded, whereas the mean number of households was included in the model. Finally, the percentage of nursing home beds in the total population was included in the model and the mean household size dropped out with a reduction of the adjusted R$^2$. This result prompted us to find a combination of the two variables that captured different aspects of a process that linked household size to the percentage of available nursing home beds. A combined index was set up (percentage of nursing home beds/mean household size) that was included in the final model with a slight but significant increase of the adjusted R$^2$. Fig 1 shows the linear relation between the model and outcome variable.

## Discussion

The hypothesis of a relationship between social connectedness and spread of COVID-19 was not confirmed by this study. The above mentioned analyses show the spread of the infection among the population aged >80 years was associated with the percentage of households comprising one member (even if it was statistically insignificant in the multivariate analyses), and inversely associated with the mean household size, the latter independently from the population age-structure and spread of COVID-19 in the general population. The spread of COVID-19 among the older adults was also independently associated with the available nursing homes beds. The model explains more than 70% of the variation of the proportion of infected patients aged >80 years.

Social relationships in Italy have changed dramatically in the last 20 years. The IStat reported that, in 2018, one third of Italian families (33%) were composed of only one person and only 5.3% had ≥ 5 members [10]. If we include couples without children (20.1%), more

than half (53.1%) of the 25 million Italian families comprise ≤2 people [4]. The 2018 ISTAT report stated that more than 25% of the population aged >75 years has no one to count on in case of need, and living alone has become the most common living arrangement, as it is all over in Europe (about one third of households comprise only one member) [11]. More than 50% of the population aged >85 years in Italy is living alone [12], and in some regions, as in Veneto–Northern Italy, this percentage increases to 75%, with about 25% of older adults that are not self-sufficient [13]. More than 50% of the residents aged >75 years in Lombardy, the most impacted region in Italy, live alone; here, in the last 10 years the household size decreased by about 10%[14], whereas in Campania (Southern Italy with one of the lowest proportions of infected patients aged >80 years) it decreased by <3% [11, 15]. It is likely that in a situation of forced isolation, social relationships represent a powerful tool to reduce the risk of contagion, allowing community-dwelling older adults to receive some assistance (bringing food or drugs home, thus facilitating the older adults to accomplish with social distancing). This kind of support is increasing in many Italian cities.

The protective power of social connectedness emerged in many crises, mainly because of the increasing prevalence of bio-psycho-social frailty among older adults who have to face repeated "environmental" stresses, such as a pandemic or a heat wave. This was not the case in nursing homes that banned visitors since the early phase of the epidemic, even though this approach did not prevent the spread of the infection. Nursing homes are dealing with the same problems as all closed communities, the struggle to ensure social distancing between people who need care, and the lack of PPE, as was the case at national level. In short, social distancing does not necessarily imply social isolation. Similarly, social connectedness does not imply physical closeness (which is dangerous in an epidemic) with social contacts. Italian nursing homes (called RSAs–translated in English as social and health care facilities) house >265,000 older adults in Italy. A very recent cross-sectional survey conducted by ISS [16] is going to explore the spread of COVID-19 in a sample of 2,556 RSAs because of several micro epidemics took place in numerous facilities. The role played by residential care facilities in Italy as well as in many other countries, due to their lack of preparedness to this kind of events, is progressively emerging [17, 18]. In some cases public policies aimed at discharging COVID-19 cases from hospitals to nursing homes in order to improve the capacity of the health systems to face the lack of hospital beds are suspected to have increased also the risk of infection among the nursing homes' hosts [19].

There are some limitations in the present study. First, the study analyzed the proportion of cases in patients aged >80 years instead of analyzing age-specific infection or mortality rates according to regions, as these data are not retrievable from public sources at this stage of the infection. Second, the quality of social connectedness parameters is limited, even though this is a limit of each study exploring social relationships based on routine data instead of gathering information ad hoc. Finally, the number of performed tests, which varies enormously according to the region (ranging from 1.8 to 18.2 per 1,000 inhabitants) [6], could affect the proportion of confirmed cases. The policy at the regional level regarding testing varies from an approach aimed at identifying asymptomatic cases as much as possible (Veneto) to one aimed at testing only the symptomatic in order to give priority to most severe cases. Of course, the most severe cases are more prevalent among the very old, so the differences in the approach could lead to an overestimation of cases in patients aged >80 years in regions where the total number of tests is lower and the Number of Confirmed Cases/Number of Performed Tests ratio (NCT/NPT*100) is higher. Indeed, this could be the case since the correlation between the NCT/NPT ratio and the proportion of cases in patients aged >80 years was robust (0.603; p = 0.004), but the multivariate model remained unchanged and the NCT/NPT ratio was not shown as a statistically significant variable.

## Conclusion

The association of social connectedness with the spread of COVID-19 among older Italian adults, hence older adult mortality rate, is not confirmed. Paradoxically, it seems that the variables associated with social isolation are risk factors for increase in the proportion of cases in Italian patients aged >80 years among the total number of cases. This is consistent with the observation that social relationships are a protective factor against increased mortality rates during a crisis impacting the frailest populations. Nursing homes bed rate is one of the determinants of SARS-CoV-2 infection rate among the individuals aged>80 in Italy.

## Author Contributions

**Conceptualization:** Giuseppe Liotta, Maria Cristina Marazzi, Stefano Orlando, Leonardo Palombi.

**Data curation:** Giuseppe Liotta, Maria Cristina Marazzi, Stefano Orlando, Leonardo Palombi.

**Formal analysis:** Giuseppe Liotta, Maria Cristina Marazzi, Stefano Orlando, Leonardo Palombi.

**Methodology:** Giuseppe Liotta, Maria Cristina Marazzi, Stefano Orlando, Leonardo Palombi.

**Writing – original draft:** Giuseppe Liotta, Maria Cristina Marazzi, Stefano Orlando, Leonardo Palombi.

**Writing – review & editing:** Giuseppe Liotta, Maria Cristina Marazzi, Stefano Orlando, Leonardo Palombi.

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
