## [Decision Letter · Decision Letter 0]

17 Apr 2020

PONE-D-20-10315

Is social connectedness a risk factor for the spreading of COVID-19 among older adults? The Italian paradox

PLOS ONE

Dear DR. LIOTTA,

Thank you for submitting your manuscript to PLOS ONE. After careful consideration, we feel that it has merit but does not fully meet PLOS ONE’s publication criteria as it currently stands. Therefore, we invite you to submit a revised version of the manuscript that addresses the points raised during the review process.

We would appreciate receiving your revised manuscript by Jun 01 2020 11:59PM. To enhance the reproducibility of your results, we recommend that if applicable you deposit your laboratory protocols in protocols.io, where a protocol can be assigned its own identifier (DOI) such that it can be cited independently in the future. For instructions see: http://journals.plos.org/plosone/s/submission-guidelines#loc-laboratory-protocols

We look forward to receiving your revised manuscript.

Kind regards,

Pasquale Abete

Academic Editor

PLOS ONE

Additional Editor Comments (if provided):

The manuscript is interesting but it merits a minor revision.

2.  To meet our reproducibility criteria, we kindly ask that all the material and methods used in your analysis are reported in more detail ; for example,  the methods section should contain a list of all the data sources used, an explanation on how data were extracted, and provide information on the statistical analysis performed. Furthermore, please specify the date range of the data here reported, and provide   a table of relevant demographic details of the patients included in the analysis.

Reviewers' comments:

Reviewer's Responses to Questions

**Comments to the Author**

1. Is the manuscript technically sound, and do the data support the conclusions?

Reviewer #1: Yes

Reviewer #2: Yes

2. Has the statistical analysis been performed appropriately and rigorously? 

Reviewer #1: Yes

Reviewer #2: Yes

3. Have the authors made all data underlying the findings in their manuscript fully available?

Reviewer #1: Yes

Reviewer #2: Yes

4. Is the manuscript presented in an intelligible fashion and written in standard English?

Reviewer #1: Yes

Reviewer #2: Yes

5. Review Comments to the Author

Reviewer #1: The study is a of great interest. The variables included in the model explains more than 70% of the variation of the proportion of infected patients aged >80 years. The idea comes from the evidence of spread of disease in family clusters and thus the extensive intergenerational contact as a possible determinant of the severity of the pandemic among the older Italian adults. Interestingly the study demonstrates the pandemic was more severe in regions with higher family fragmentation and increased availability of residential health facilities. The Nursing Home experience need a more accurate description. I guess that the negative experience of Lombardy, where with a regional law (DELIBERAZIONE N° XI / 2906 Seduta del 08/03/2020) RSA were dedicated to low intensity care for COVID patients, should be cited and discussed as potential cause of COVID spreading.

Reviewer #2: The authors analyzed the percentage of infected patients aged >80 years, available nursing home beds, COVID-19 incidence rate, and the number of days from when the number of positive tests exceeded 50 (epidemic maturity). They included in the analysis mean household size and percentage of households comprising one person, in the region. The authors found that paradoxically, the results are opposite of what was previously reported. They concluded that the pandemic was more severe in regions with higher family fragmentation and increased availability of residential health facilities.

The manuscript is very interesting and topic. I have only a concern. do you have some data about COVID-related letality and/or mortality?

6. PLOS authors have the option to publish the peer review history of their article (what does this mean?). If published, this will include your full peer review and any attached files.

Reviewer #1: Yes: Francesco CACCIATORE

Reviewer #2: No

---

## [Author Response · Author response to Decision Letter 0]

29 Apr 2020

Reviewer 1

thank you for your answer. We now mention in the discussion the regional law you indicated with a short comment

Reviewer 2

Thank for your answer. There are data about COVID-19 related mortality and letality, however they are not yet provided as age-specific rate at regional level. This prevented us from any further analyses, since the paper is based on data fully available to the public. However, there are growing evidences of the dramatic impact on nursing home residents survival (see https://www.epicentro.iss.it/coronavirus/pdf/sars-cov-2-survey-rsa-rapporto-2.pdf also cited in the paper).

---

## [Decision Letter · Decision Letter 1]

5 May 2020

Is social connectedness a risk factor for the spreading of COVID-19 among older adults? The Italian paradox

PONE-D-20-10315R1

Dear Dr. LIOTTA,

We are pleased to inform you that your manuscript has been judged scientifically suitable for publication and will be formally accepted for publication once it complies with all outstanding technical requirements.

With kind regards,

Pasquale Abete

Academic Editor

PLOS ONE

Additional Editor Comments (optional):

No comments.

Reviewers' comments:

Reviewer's Responses to Questions

**Comments to the Author**

1. If the authors have adequately addressed your comments raised in a previous round of review and you feel that this manuscript is now acceptable for publication, you may indicate that here to bypass the “Comments to the Author” section, enter your conflict of interest statement in the “Confidential to Editor” section, and submit your "Accept" recommendation.

Reviewer #1: All comments have been addressed

Reviewer #2: All comments have been addressed

2. Is the manuscript technically sound, and do the data support the conclusions?

Reviewer #1: Yes

Reviewer #2: Yes

3. Has the statistical analysis been performed appropriately and rigorously? 

Reviewer #1: Yes

Reviewer #2: Yes

4. Have the authors made all data underlying the findings in their manuscript fully available?

Reviewer #1: Yes

Reviewer #2: Yes

5. Is the manuscript presented in an intelligible fashion and written in standard English?

Reviewer #1: Yes

Reviewer #2: Yes

6. Review Comments to the Author

Reviewer #1: I have no more suggestions to improve your manuscript. The paper is suitable for publication in present form

Reviewer #2: The authors have been addressed all questions arised. The manuscript may be acceptable to be published in PONE.

7. PLOS authors have the option to publish the peer review history of their article (what does this mean?). If published, this will include your full peer review and any attached files.

Reviewer #1: No

Reviewer #2: No

---

## [Editor Report · Acceptance letter]

8 May 2020

PONE-D-20-10315R1 

Is social connectedness a risk factor for the spreading of COVID-19 among older adults? The Italian paradox 

Dear Dr. liotta:

I am pleased to inform you that your manuscript has been deemed suitable for publication in PLOS ONE. Congratulations! Your manuscript is now with our production department. 

With kind regards,

on behalf of

Prof. Pasquale Abete 

Academic Editor

PLOS ONE